# Direct Fabrication of Ultra-Sensitive Humidity Sensor Based on Hair-Like Laser-Induced Graphene Patterns

**DOI:** 10.3390/mi11050476

**Published:** 2020-04-30

**Authors:** Jun-Uk Lee, Yong-Won Ma, Sung-Yeob Jeong, Bo-Sung Shin

**Affiliations:** 1Department of Cogno Mechatronics Engineering, Pusan National University, Busan 46241, Korea; lju3534@naver.com; 2Interdisciplinary Department for Advanced Innovative Manufacturing Engineering, Pusan National University, Busan 46241, Korea; decentsoul@naver.com (Y.-W.M.); ysjsykj8025@naver.com (S.-Y.J.); 3Department of Optics and Mechatronics Engineering, Pusan National University, Busan 46241, Korea

**Keywords:** 3-D porous structures of graphene (3-D PGS), laser induced graphene (LIG), laser direct writing (LDW), hair-like, humidity sensor, polyimide (PI)

## Abstract

Three-dimensional (3-D) porous graphitic structures have great potential for sensing applications due to their conductive carbon networks and large surface area. In this work, we present a method for facile fabrication of hair-like laser induced graphene (LIG) patterns using a laser scribing system equipped with a 355 nm pulsed laser. The polyimide (PI) film was positioned on a defocused plane and irradiated at a slow scanning speed using a misaligned laser beam. These patterns have the advantages of a large surface area and abundant oxidation groups. We have applied the hair-like LIG patterns to a humidity sensor. The humidity sensor showed good sensitivity characteristics and a large amount of electronic carriers can be stored.

## 1. Introduction

Graphene is a two-dimensional (2-D) monolayer of $sp^{2}$ carbon atoms arranged in a honeycomb lattice [1]. Due to its structure, graphene has unique physical and chemical properties. Although it has great potential for sensing applications, the 2-D structure of graphene is not easy to apply to sensors due to van der Waals forces. These cause aggregation, restacking, and defects in the graphene sheets that result in dramatically reduced surface area and uneven dispersion in solvents [2]. Recently, 3-D porous structures of graphene (3-D PGS) have been reported to have better properties than 2-D graphene [3,4,5,6]. Because of their extraordinary performances, 3-D PGS have been widely used in a variety of applications such as touch sensors [7], strain sensors [8,9], solar cells [10,11], biosensors [12,13,14,15], and supercapacitor [16,17,18,19]. Fabrication methods for making 3-D PGS include chemical vapor deposition (CVD) on a template and a drying method using graphene hydrogel. These methods give less control over the formation of 3-D structures and have the disadvantages of requiring high-temperature conditions, long synthesis processes, and high cost [4,5,6]. For graphene sensor applications, various facile methods for fabrication of 3-D PGS based on a sensor platform are imperative for realization of better sensor applications. In 2014, Tour et al. revealed a method by which they could easily fabricate a LIG electrode of 3-D PGS using a CO_2_ infrared laser to irradiate directly a commercial PI film [20]. LIG has good properties such as large surface area (~340 m^2^/g), high thermal stability (>900 °C), and excellent conductivity (5–25 S/cm). Therefore, the fabrication of sensors using LIG is very simple and cost-effective, showing many advantages over lithographic sensors [20]. This is very important because the resulting laser-patterned porous and conductive films have specific structures and various functional groups in a flexible substrate that can be applied to numerous applications such as physical sensor [21,22], gas sensor [23,24], supercapacitor [25,26] and bio sensor [27,28].

In this paper, we present an approach for fabrication of a porous, conductive, 3-D carbon network obtained by direct laser writing onto PI sheets. This is done under ambient conditions with a cost-effective laser scribing system equipped with a 355 nm pulsed laser. Due to the wavelength characteristic of 355 nm pulsed laser, it is effective to manufacture LIG by irradiating the polyimide with a laser. By controlling the alignment, laser speed and focal plane, we could control changes in the LIG pattern’s morphology and chemistry. The most important feature of our LIG manufacturing method is the condition of the laser. We used misaligned and defocused beams when irradiating the PI with a laser. In order to increase the overlap rate of the pulse, the scanning speed of the laser was adjusted to 10 mm/s, and hair-like LIG patterns were made under the conditions mentioned. These patterns have the advantage of having a large surface area and rich functional groups. This results in improving the characteristics of the sensor. Therefore, we propose a new approach to fabricate a humidity sensor in a simple one-step process without the complex fabrication processes used for conventional humidity sensors [29,30,31]. We propose for the first time to create a humidity sensor using the hair-like LIG patterns.

## 2. Experimental

### 2.1. Laser System

In this experiment, LIG patterns were fabricated using a laser scribing system equipped with a 355 nm pulsed laser (Series 3500 UV Laser from DPSS Laser, Inc, Santa Clara, CA, USA). A photograph of the laser system configuration is shown in Figure 1, and the specifications of the laser are as shown in Table 1. We used a computational method with 2-D CAD (AutoCAD, San Rafael, CA, USA). For a one-step fabrication process, we used the Laser Direct Writing (LDW) method with a Galvano scanner (Model 6230 Galvanometer Optical Scanner from Cambridge Technology, Inc., Bedford, MA, USA). LDW method has many advantages including short process time, less sample damage (no contact), and the possibility of making a variety of patterns. And an F-θ lens (S4LFT4100/075 Telecentric Scan Lens from Sill Optics, Wendelstein, Germany) was used to radiate the laser light vertically and evenly over the material surface.

### 2.2. Principle of LIG Using a 355 nm Pulsed Laser

PI thin film has a very high absorption rate with respect to 355 nm wavelength [32]. Although the absorption rate varies depending on the fluence of the laser, it has an absorption rate of over 85% from very low fluence of $10^{-3}$ J/${\mathrm{cm}}^{2}$. The threshold fluence of a polyimide for a 355 nm laser is known to be 0.1 J/${\mathrm{cm}}^{2}$ [33]. If the film is irradiated by a laser with fluence above this threshold, the morphological and chemical changes of the surface are accompanied by a chemical reaction with atmospheric nitrogen and oxygen, increasing the C/O and C/N ratios [34]. Recently, the Tour team found that LIG based on heat treatment of polyimide was fabricated by irradiating polyimide with a ${\mathrm{CO}}_{2}$ continuous wave (CW) laser (wavelength 10.640 µm) [20]. When the polyimide film was subjected to a high pressure (~3 GPa) and high temperature (2400 K or more), the hexagonal ring structure of the polyimide was broken and layered graphene clusters crystallized from polyimide without the need for metal catalysts. Most of the carbon again formed carbon dimers due to the high pressure. Non-carbon atoms such as H, N, and O escaped from the polyimide to form $H_{2}$, $H_{2}O$, $N_{2}$, and ${\mathrm{NH}}_{3}$. Carbon rings formed and together made up a graphene layer [35]. Therefore, 3-D PGS is formed where the gases left.

LIG, described above, is based on photothermal effect caused by a ${\mathrm{CO}}_{2}$ CW laser. However, in the case of 355 nm pulsed laser processing, photochemical effect as well as photothermal effect occur. In the polyimide structure, the C-N energy bond (shown in Table 2) is the only one lower than 3.5 eV (energy of a 355 nm wavelength photon). When a 355 nm pulsed laser with moderate power is irradiated to the PI, carbonyl groups are removed. After that, LIG is formed in the process of recovering while forming the structures among the carbons.

Figure 2 shows the schematic diagram of the LIG formation process of a 355 nm pulse laser. We maximize the LIG formation by increasing the overlap rate of the laser beam.

### 2.3. Fabrication of the Hair-Like LIG Patterns

Commercial PI film (DUPONT™ KAPTON^®^ HN, Wilmington, DE, USA) with a nominal thickness of 125 µm was irradiated at defocused plane (+1.5 mm) and misaligned laser beam. Figure 3a shows the defocused beam condition. Since the area where the beam is focused is separated from the focused plane by +1.5, the spot size of the laser increases.

Figure 3b is an illustration of the power intensity distribution of a laser pulse. The laser intensity was adjusted to become stronger as it was directed downward. It also shows an illustration of when the pulses overlap. Figure 3c–j shows FE-SEM images of line pattern fabricated with the laser speed of 90, 80, 70, 60, 50, 40, 30 and 20 mm/s. When misaligned and defocused beams are irradiated to the PI, these pictures show morphological characteristics according to speed. When the laser speed became 10 mm/s, it was observed that LIG flakes were formed in the line pattern as shown in Figure 3k and the PI film itself is removed in regions with a strong laser intensity distribution. If an aligned beam is used, the LIG flakes do not form on the surface and the area irradiated with the laser are uniformly removed. This causes just a decrease in the surface area, which reduces the sensitivity of the sensor. Our goal is to fabricate a sensitive sensor platform with a large surface area. Therefore, hair-like LIG patterns were fabricated by multi-patterning at the speed of 10 mm/s and the conditions mentioned above (misaligned and defocused laser beam).

Figure 4 shows the fabrication and morphological characteristics of hair-like LIG patterns. As shown in Figure 4a, the diameter of the laser spot is about 100um when the laser is defocused by +1.5 mm. Because the laser speed is 10 mm/s, the overlap factor ($O_{f}$) of the pulse was calculated to be 99.66% ($O_{f}=\left(1-\frac{v/f}{D+vt}\right)\times 100$: Where $O_{f}$ is the overlapping factor, D is the laser beam diameter on the work piece, t the pulse duration (ms), f the pulse frequency (Hz), and v refers to the laser speed (mm/s)). Due to the high heat and pressure caused by pulse overlapping, the laser scribed portion is removed. The blue line shows the laser path. In this case, a unidirectional scanning strategy was used for one-step processing. Through this, we can fabricate hair-like LIG patterns uniformly. Figure 4b,c shows actual pictures of the fabricated hair-like and LIG patterns. Through the misaligned laser beam irradiation, multiple LIG flakes are gathered to one side shown in Figure 4c. The morphological advantages of these hair-like LIG patterns are airy and have a large surface area due to the features of the hair-like structure. Due to this, it can be applied as a sensitive humidity sensor without stopping the gas flow. Figure 3d,e are photos of hair-like LIG patterns in banding. Figure 4f is an optical microscope photograph. A reflection phenomenon was observed in the LIG patterns.

## 3. Results and Discussion

### 3.1. The Morphological and Chemical Characteristics of Hair Like LIG Patterns

We observed the formation of 3-D PGS in hair-like LIG patterns through FE-SEM pictures and predicted the chemical structure through Raman spectra analysis.

Figure 5a shows a cross sections of the hair-like LIG pattern. The thickness of this pattern is 140 µm, similar to the thickness of human hair. In Figure 5b,c, FE-SEM images show micro-scale graphene-like flakes and the surface of the flakes comprise a myriad of nano-sized LIG particles. Figure 5e,f shows the morphological characteristics inside the hair-like LIG patterns. It shows a porous structure. To confirm the crystal and electronic structure of the newly fabricated hair-like LIG patterns, a Raman (NRS-5100, Easton, MD, USA) scattering measurement was performed. The Raman spectra in Figure 5d shows three distinct peaks: a D peak near 1350 cm^−1,^ a G peak near 1580 cm^−1^, and a 2-D peak near 2700 cm^−1^. The layer number identification of graphene can be confirmed by the $I_{2D}$/$I_{G}$ ratio (The higher the $I_{2D}$/$I_{G}$ ratio, the fewer layers of graphene). The $I_{2D}$/$I_{G}$ ratio of the hair-like LIG patterns was calculated to be 0.51. The layer of the graphene was expected to be more than 10 stacked graphene layers. The $I_{D}$/$I_{G}$ ratio indicates graphene crystallinity (the lower the ratio is, the greater the crystallinity). The $I_{D}$/$I_{G}$ ratio of the LIG patterns was 1.18. LIG patterns have many defects. Most of these defects can be predicted by the oxidation group.

### 3.2. Application of the Hair-Like LIG Patterns for Humidity Sensor

Because the multi-layered graphene with porous structure has many defects, such as abundant oxygen groups, these LIG patterns have the possibility of excellent sensing ability.

The facile fabrication of a hair-like humidity sensor is shown Figure 6a. First of all, we designed the patterns using 2-D CAD. The laser was irradiated to the PI at 100 µm intervals in a square size of 1 × 1 cm. Then, silver paste was applied to both sides of the patterns to use as an electrode. We did experiments in the water vapor detection box (as shown in Figure 6b). When water vapor was released through a humidifier, it passed the humidity sensor and then was quickly removed by fan motors. Figure 6c demonstrates the I-V characteristics of the sensors with a voltage range of −10 to 10 V. These electrical characteristics were measured by the source meter (KEITHLEY, Source Meter 2450, Tektronix Co., Solon, OH, USA). The graph has showed difference between LIG patterns and PI. PI shows a lot of noise, but LIG patterns show stable electrical properties. Through this, it can be predicted that a conductive carbon network is well formed. Figure 6d shows the hygroscopic mechanism of the humidity sensor. Due to polarity of water molecules, intermolecular force is generated between LIG patterns and water molecules. Therefore, water molecules are adsorbed to the surface of the LIG pattern by van der Waals and surface tension. Absorption processes of water molecules are shown in Figure 6e. There are three adsorption processes: chemical absorption, physical adsorption, and condensation [36]. The first is a chemisorbed state, where water molecules chemically bind to the oxygen group and at the same time $H_{2}O$ dissociates to form hydroxyl groups. The second is the physisorbed state. As the amount of water increases, other water molecules and oxygens interact with the chemisorbed water molecules and are maintained by the surface force. This continues to sustain a multimolecular layer of water. As the multilayer physical adsorption progresses, the physisorbed water can be ionized under an electrostatic field to produce a large number of hydronium ions ($H_{3}O^{+}$) as charge carriers. With further increase in humidity, the physisorbed water layers gradually exhibit liquid-like behavior (condensation). In bulk liquid, proton hopping between adjacent water molecules occurs in the surface, with charge transport taking place via the conductivity generated by a Grotthuss chain reaction ($H_{2}O+H_{3}O^{+}\to H_{3}O^{+}+H_{2}O)$ conductivity [37].

### 3.3. Electrical Characteristics of Hair-Like LIG Humidity Sensor

We conducted several experiments to check the electrical characteristics of the humidity sensor.

Figure 7a demonstrates the capacitance response of the humidity. With the fluctuation range from 0 to 20, capacitance changes for every 10 sections of RH are clearly distinguished. It can be regarded as a more sensitive capacitance humidity sensor than a resistance humidity sensor with a variation width less than 10 [31]. When RH is less than 50%, the change in capacitance is small, but when RH is 50 or more, the change is large. We predict that when RH is more than 50%, the multilayer physical absorption rate of moisture increases, so that many moisture molecules dissociate into hydronium ions, resulting in an increase in capacitance change. The hysteresis test (Figure 7b) was performed by changing relative humidity from 30% to 90%. It is good to have small deviation from absorption to desorption. This graph shows a bit more linearity when moisture is desorbed. Figure 7c presents the transient response of the humidity sensor during cycling between 30% and 90% RH. Through this experiment, we were able to confirm the electrical stability of the humidity sensor of LIG patterns like hair. And When the RH is 30 and 90, the fluctuation range is about 45 times. This is a very large value compared to a few percent increase [29]. As shown in Figure 7d, the response and recovery time were calculated by conducting a moisture saturation experiment of the sensor. The response time and recovery time were 8 and 10 s, respectively. The reaction time and recovery time, which took several tens of hundreds of seconds [30], were reduced to several seconds. And when moisture is saturated, the capacitance rises to 4 nF or more. This is about 133,333 × 0.3 pF at 30%. From these results, we were able to confirm that the sensor can hold many electronic carriers such as hydronium ions. This is due to the rich oxidation groups distributed over a large surface area.

## 4. Conclusions

To fabricate a hair-like LIG patterns, we placed the PI film on a defocused plane and slowly irradiated the laser with a misaligned beam. We confirmed that FE-SEM and Raman spectra showed that hair-like LIG patterns had a porous structure and had abundant oxidation groups. We applied it as a humidity sensor to confirm the applicability of these patterns. For simplification of the process, the pattern was designed with 2-CAD and fabricated using LDW method. This hair-like humidity sensor showed good sensitivity and was also excellent in storage capacity for large quantities of electrical carriers. Therefore, our new form of sensor platform presents excellent characteristics which have potential for fundamental and practical work.

## Figures and Tables

**Figure 1 micromachines-11-00476-f001:**
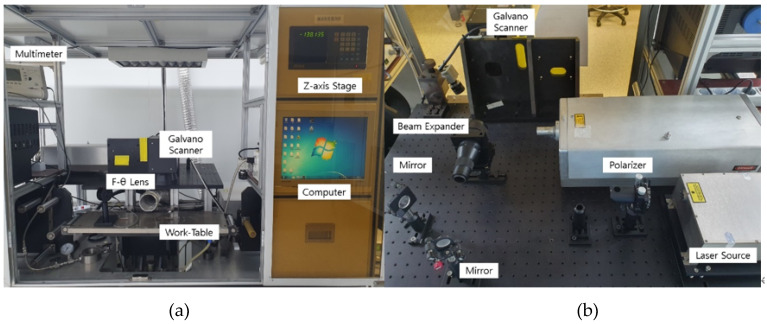
Laser system: (**a**) laser set-up; (**b**) optical system.

**Figure 2 micromachines-11-00476-f002:**
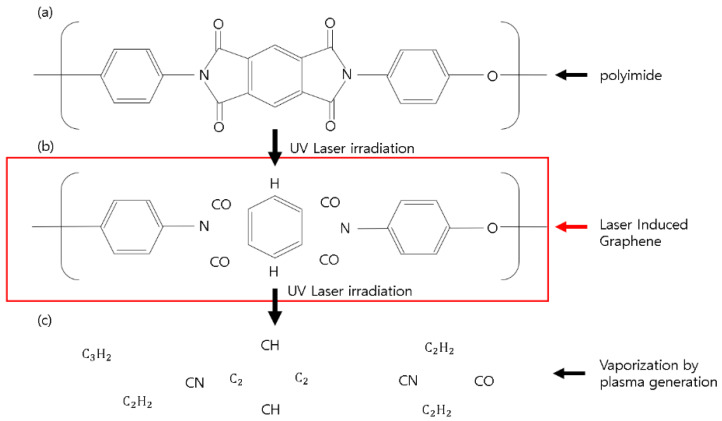
Principle of laser induced graphene (LIG) using a 355 nm pulsed laser: (**a**) polyimide structure, (**b**) structural change of polyimide after laser irradiation, and (**c**) vaporization by plasma generation.

**Figure 3 micromachines-11-00476-f003:**
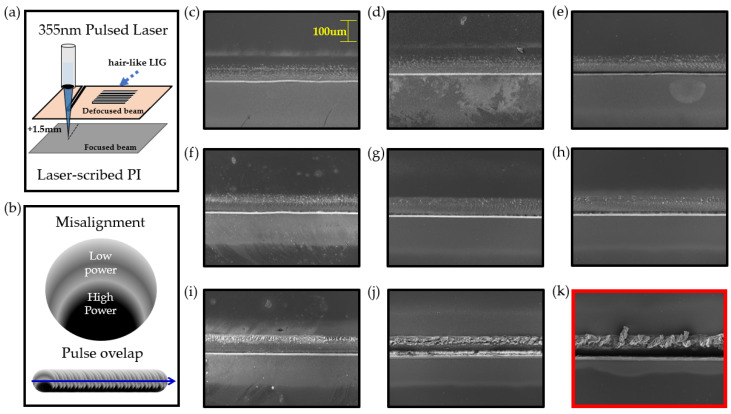
(**a**) Schematic illustration of 355 nm pulsed laser condition (defocused beam), (**b**) illustration of the power intensity distribution of a laser pulse, (**c**–**j**) FE-SEM images of line pattern fabricated with the laser speed of 90, 80, 70, 60, 50, 40, 30 and 20mm/s, (**k**) FE-SEM image of line pattern with LIG flakes fabricated at 10 mm/s.

**Figure 4 micromachines-11-00476-f004:**
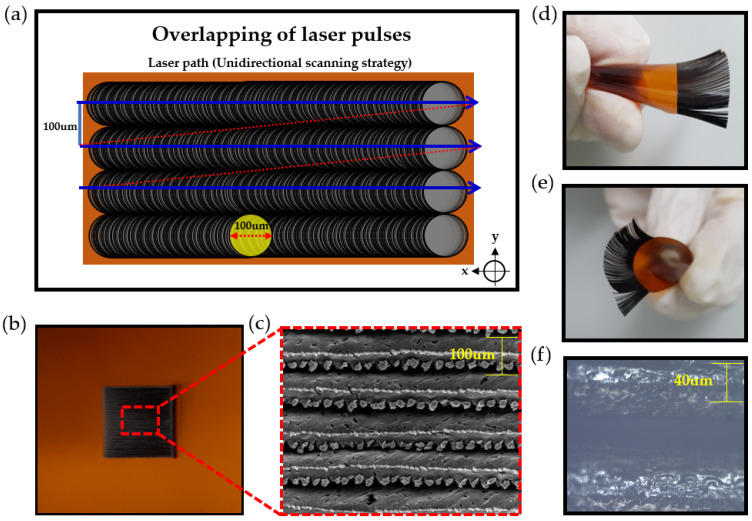
Fabrication and morphological characteristics of hair-like LIG patterns. (**a**) Schematic illustration of pulse overlap; (**b**) photograph of hair-like LIG patterns; (**c**) FE-SEM image of hair-like LIG patterns; (**d**,**e**) photograph of hair-like LIG patterns in a curved state; and (**f**) optical image of LIG patterns.

**Figure 5 micromachines-11-00476-f005:**
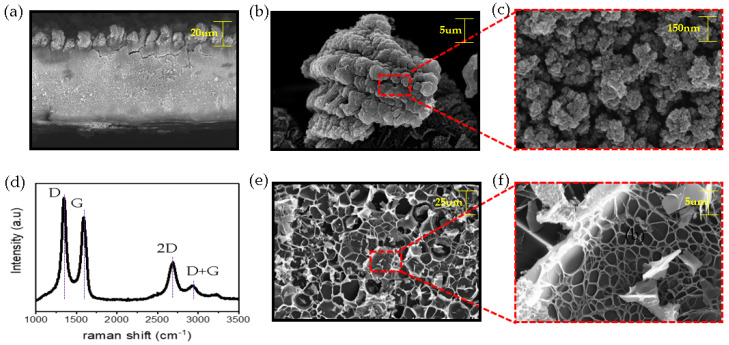
Three-dimensional porous structures of graphene (3D-PGS) properties of hair-like LIG patterns: (**a**) cross-section FE-SEM image, (**b**,**c**) FE-SEM images of hair-like LIG patterns, (**d**) Raman spectra of LIG patterns in the range 1000–3500 ${\mathrm{cm}}^{-1}$, (**e**,**f**) FE-SEM images inside hair-like LIG patterns.

**Figure 6 micromachines-11-00476-f006:**
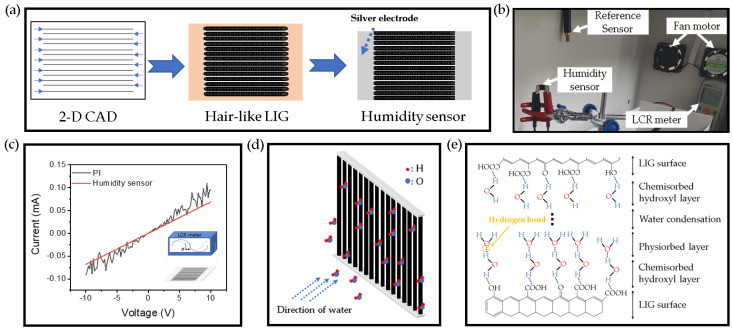
(**a**) Method for fabrication of a hair-like LIG humidity sensor, (**b**) photograph of the humidity measurement set-up, (**c**) I-V characteristics of the humidity sensor, (**d**) schematic illustration of a humidity sensor exposed to humidity, (**e**) absorption processes of water molecules by hydrogen bonding to the LIG surface after moisture exposure.

**Figure 7 micromachines-11-00476-f007:**
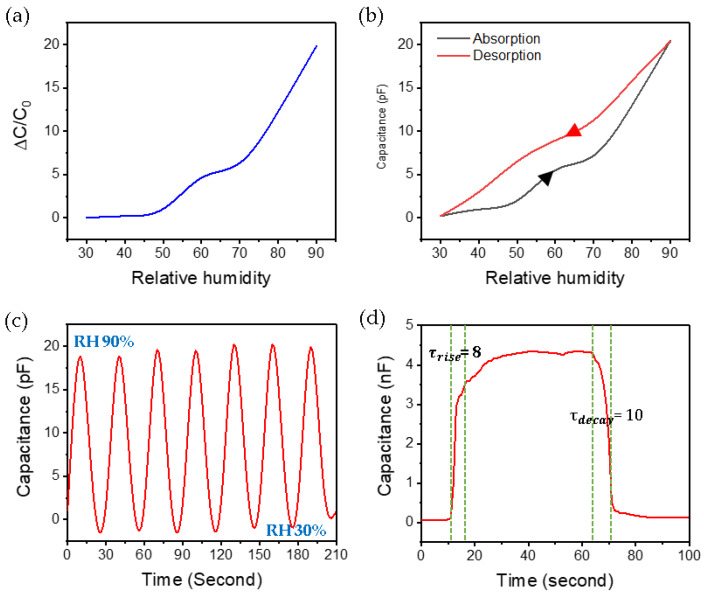
(**a**) Measurement of capacitance change according to relative humidity 30% to 90%, (**b**) hysteresis characteristics of the humidity sensor, (**c**) cycling stability test of the humidity sensor, (**d**) capacitance-time curve of the sensor from 30% to moisture saturation.

**Table 1 micromachines-11-00476-t001:** Specifications of the 355 nm UV pulsed laser.

Parameter	Unit	Value
Wavelength	nm	355
Average power	Watt	~2.5
Pulse duration	ns	25
Repetition rate	kHz	30
Mode	-	TEM_00_
Beam diameter	mm	1.5
Beam divergence	mard	<0.5

**Table 2 micromachines-11-00476-t002:** Chemical bond energies for polymers.

Polymer Bond	C-N	C-H	C≡C	O-O	C=C	C-C	N-N	H-H
Bond energy (eV)	3.04	4.30	8.44	5.12	6.40	3.62	9.76	4.48

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
