# Peer review of "Direct Fabrication of Ultra-Sensitive Humidity Sensor Based on Hair-Like Laser-Induced Graphene Patterns"

_micromachines, 2020, doi:10.3390/mi11050476_

Round 1

Reviewer 1 Report

  1. The last paragraph in the Introduction of the paper between lines 45 and 52 in which the authors mention about the novelty and originality of their work must be modified to make it more specific about the exact purpose of the work and its novelty as compared to other works mentioned in the references list of the paper. The authors should be more specific about the novelty of their work and mention explicitly and with more details what the new things are that they did at the present work as compared to other similar works in the literature.
  2. How do the authors define the "absorption rate" which they mention about the polyimide in lines 67-69?
  3. The reasons for using a defocused and misaligned laser beam for the fabrication of the hair-like LIG described in Section 2.3, are not clear. The authors should provide with more explanation as to why they used a misaligned laser beam. Obviously the defocused beam provides with a larger irradiated area - what was the area of the defocused beam on the PI film? The misalignment of the beam obviously has as a result an inhomogeneous laser fluence around the beam area on the substrate but how that exactly lead to the formation of the hair-like patterns?
  4. The results from the sensor characterization and its characteristics (Figure 6 and Section 3.3) should be compared and discussed with the results for similar sensors such as for instance those reported in References 29-31 of the paper.
  5. I am not able to access the Supplementary Material of the paper - link indicated in line 198: www.mdpi.com/xxx/s1

Author Response

Hello? Thank you for reviewing my first paper.

1.The last paragraph in the Introduction of the paper between lines 45 and 52 in which the authors mention about the novelty and originality of their work must be modified to make it more specific about the exact purpose of the work and its novelty as compared to other works mentioned in the references list of the paper. The authors should be more specific about the novelty of their work and mention explicitly and with more details what the new things are that they did at the present work as compared to other similar works in the literature.

We modified the introduction to clearly mention the novelty and orginality of our experiment. The method of forming LIG like hair through the conditions of laser and the application to humidity sensor due to the morphological and chemical advantages of patterns are described in detail.

2.How do the authors define the "absorption rate" which they mention about the polyimide in lines 67-69?

Sorry for the mistake. This part has been deleted.

3.The reasons for using a defocused and misaligned laser beam for the fabrication of the hair-like LIG described in Section 2.3, are not clear. The authors should provide with more explanation as to why they used a misaligned laser beam. Obviously the defocused beam provides with a larger irradiated area - what was the area of the defocused beam on the PI film? The misalignment of the beam obviously has as a result an inhomogeneous laser fluence around the beam area on the substrate but how that exactly lead to the formation of the hair-like patterns?

When the laser is defocused by +1.5mm, the diameter of the laser beam is about 100um. And because the laser speed is 10mm / s, the overlap factor  of the pulse was calculated to be 99.66%. Due to the high heat and pressure caused by pulse overlapping, the laser scribed portion is removed. The beam of the pulse misalignment was misaligned so that the energy is stronger as the laser beam is directed upward. This is because LIG flakes are formed on the surface when the beam is strengthened to one side. 

4.The results from the sensor characterization and its characteristics (Figure 6 and Section 3.3) should be compared and discussed with the results for similar sensors such as for instance those reported in References 29-31 of the paper.

As you mentioned, the references in [29-31] are compared in Figures 6 (a), (c) and (d). Thank you for pointing out.

5.I am not able to access the Supplementary Material of the paper - link indicated in line 198: www.mdpi.com/xxx/s1

I am not sure why this part is not accessible. I'll ask the editor.

Best regards,

JUNUK LEE

Reviewer 2 Report

The manuscript (#micromachines_784096) describes a LIG based humidity sensor on polyimide. After carefully reading the manuscript, my comments are as follows:

  1. The authors did not mention the advantage of “hair-like” structure for humidity sensor. It seems to me that hair-like structure was obtained by their process without any design method. If so, I suggest that they should remove “hair-like” in the title of the manuscript.
  2. In Table 1, the beam diameter was 0.1 mm. Was it the diameter before f-theta lens or after the lens? It seems the diameter listed was in the focus plane.
  3. Section 2.3 “Fabrication of the hair-like LIG patterns,” the unit of the fluence was J/mm^2. Was it a typo in the unit? Also, the unit should be the same as described in Section 2.2, which was J/cm^2. Also, the description of the misaligned process, either described in the context or shown in Figure 3, was not clear.
  4. Are the results of electrical characteristics repeatable?
  5. There is no Figure 6 (e) in the manuscript. Please correct the description in line 156 on Page 6..

Author Response

Hi Thank you very much for reviewing my first paper.

1.The authors did not mention the advantage of “hair-like” structure for humidity sensor. It seems to me that hair-like structure was obtained by their process without any design method. If so, I suggest that they should remove “hair-like” in the title of the manuscript.

In part 2.3, the features of the hair-like structure for figure d were described. And in Figure 4 (a), we compared it with the thickness of human hair.

The morphological advantages of these hair-like LIG patterns are airy and have a large surface area due to the features of the hair-like structure. Due to this, it can be applied as a sensitive humidity sensor without stopping the gas flow.

The thickness of this pattern is 140um, similar to the thickness of human hair.

2.In Table 1, the beam diameter was 0.1 mm. Was it the diameter before f-theta lens or after the lens? It seems the diameter listed was in the focus plane.

The laser we used is DPSS LASER INC, Series 3500 UV LASER. The diameter of the laser beam of this laser itself is 1.5 mm. This is the value before entering the f-theta lens, and the table data has been modified.

3.Section 2.3 “Fabrication of the hair-like LIG patterns,” the unit of the fluence was J/mm^2. Was it a typo in the unit? Also, the unit should be the same as described in Section 2.2, which was J/cm^2. Also, the description of the misaligned process, either described in the context or shown in Figure 3, was not clear.

There was a mistake in 99 line. I had miscalculated fluence. We corrected the fluence value and mm ^ 2-> cm ^ 2.

4.Are the results of electrical characteristics repeatable?

Figure 6.c is a repeated experiment for adsorption and desorption of moisture. Other experiments were repeatable.

5.There is no Figure 6 (e) in the manuscript. Please correct the description in line 156 on Page 6..

Thanks for finding the error. Fixed 6 to 5.

BEST Regards
JUNUK LEE

Reviewer 3 Report

The authors introduce the subject saying that porous graphene structure is interesting for various applications around sensors.
They take advantage that Prof. Tour revealed a method using CO2 laser irradiating a polyimide film.
Here, they use a UV laser probably with ns pulses.
The conditions used for producing the hair-like fibers are well described.
The characterization of the matter produced is well explicated.
I do not understand exactly why they use a defocused beam and not a lens with a weaker numerical aperture, if the intensity should be lower. By the way, I understand defocusing but not misalignment. and why there is an asymmetry of grapheme production.
So, then the variation of the hair-like electric properties with humidity is a kind of demonstrator of a sensor. Can we deduce from these results that it can be applied to detect other molecules? What kind of molecules?
another question is what is the repeatability of the sensors? easy desorption?

small typo
in legend fig3: (d) is missing
on line 131, there is « e » alone before the ratio.

Author Response

Hello? Thank you very much for reviewing my first paper.

As you said we started from the method used by the tour professor (CO2 laser irradiating a polyimide film). We used UV pulsed nm lasers instead of CO2 lasers.

1. why they use a defocused beam and not a lens with a weaker numerical aperture, if the intensity should be lower. By the way, I understand defocusing but not misalignment. and why there is an asymmetry of grapheme production.

The reason for using the defocused beam was to increase the area of ​​the beam and increase the pulse overlap rate. Since the NA of the experimental system was already fixed, the work stage was adjusted. Without the misaligned beam, there were no graphene flakes on the surface. This causes a reduction in surface area and poor sensor characteristics.

2.So, then the variation of the hair-like electric properties with humidity is a kind of demonstrator of a sensor. Can we deduce from these results that it can be applied to detect other molecules? What kind of molecules?

I think it's a very good question. In fact, any substance with polarity can be measured instead of water molecules, but only humidity is measured in this experiment. Since the sensor changes are observed through the measurement of capacitance,  It is predicted that polarized molecules such as Co2 and NH3 will also be measured.  Due to the lack of test equipment and materials, this part could not be tested.

3.another question is what is the repeatability of the sensors? easy desorption?

In Figure 6. (c), we conducted repeated experiments of adsorption and desorption of moisture. Other experimental results also showed repeatability. Figure 6. (d) shows fast desorption speed (short recovery time).

Best regards,

JUNUK LEE

Round 2

Reviewer 1 Report

The authors provided with satisfactory answers to the comments raised and revised their paper accordingly. The paper is now ready to be published.

Author Response

Thank you very much for your comments.

Reviewer 2 Report

  1. In Table 1, the parameter name, “pulse length,” should be changed to pulse “duration” in order to have the same name described in the formula of th overlap factor.
  2. How much power of laser was used to obtain the fluence of 1.19 J/cm^2 with the 10 mm/s laser scanning speed?
  3. Is the fluence described in Line 76, 10^(-3) mJ/cm^2, a typo in unit? Should it be 10^(-3) J/cm^2?
  4. The misaligned laser beam is a key feature of the manuscript. However, in Section 2.3 “Fabrication of the hair-like LIG patterns,” the description of the misaligned process is still not clear. Where was the laser beam supposed to align? What was high power area shown in Figure 3(a)? Was it a laser spot? The description of the misalignment in Line 118 was confusing. Why was the energy stronger when the laser beam was directed upward?

  1. The caption of Figure 3(d) was missing.

  1. The caption of Figure 4(d) was provided twice. It should be corrected.

  1. In Section 3.2, all “Figure 6” mentioned in the paragraph should be replaced by “Figure 5.”

  1. In Figure 6(c), it seems the capacitance was down to negative values. Please explain.

Author Response

Hello and thanks for the review.

  1. In Table 1, the parameter name, “pulse length,” should be changed to pulse “duration” in order to have the same name described in the formula of th overlap factor.

I modified it as you said.

  1. How much power of laser was used to obtain the fluence of 1.19 J/cm^2 with the 10 mm/s laser scanning speed?

We will continue to make mistakes in our calculations, so we will remove them.

  1. Is the fluence described in Line 76, 10^(-3) mJ/cm^2, a typo in unit? Should it be 10^(-3) J/cm^2?

It was a typo. Thanks for finding the error. Modified to J / cm ^ 2.

  1. The misaligned laser beam is a key feature of the manuscript. However, in Section 2.3 “Fabrication of the hair-like LIG patterns,” the description of the misaligned process is still not clear. Where was the laser beam supposed to align? What was high power area shown in Figure 3(a)? Was it a laser spot? The description of the misalignment in Line 118 was confusing. Why was the energy stronger when the laser beam was directed upward?

Figure 3 (a) has been completely modified. In addition, patterning FE-SEM images according to speed were attached to help understanding. And I have written clearly for you to understand.

  1. The caption of Figure 3(d) was missing.

The caption was rewritten. Thanks for finding the error.

  1. The caption of Figure 4(d) was provided twice. It should be corrected.

I removed one of the things you mentioned

  1. In Section 3.2, all “Figure 6” mentioned in the paragraph should be replaced by “Figure 5.”

Corrected the caption number again.

  1. In Figure 6(c), it seems the capacitance was down to negative values. Please explain.

The quality of the paper seems to increase as I correct what you mentioned. Thanks for the review.

In the origin program, I did a graph fitting that connects the dots, and the result was the same. I think it was because of the wide spacing of the dots.

Best regards,

JUNUK LEE